# The Parameterized Complexity of Cascading Portfolio Scheduling

**Eduard Eiben**
Royal Holloway
University of London
Department of CS
UK

**Robert Ganian**
TU Wien
Algorithms and
Complexity Group
Austria

**Iyad Kanj**
DePaul University
School of Computing
Chicago
USA

**Stefan Szeider**
TU Wien
Algorithms and
Complexity Group
Austria

## Abstract

Cascading portfolio scheduling is a static algorithm selection strategy which uses a sample of test instances to compute an optimal ordering (a cascading schedule) of a portfolio of available algorithms. The algorithms are then applied to each future instance according to this cascading schedule, until some algorithm in the schedule succeeds. Cascading scheduling has proven to be effective in several applications, including QBF solving and generation of ImageNet classification models.

It is known that the computation of an optimal cascading schedule in the offline phase is NP-hard. In this paper we study the parameterized complexity of this problem and establish its fixed-parameter tractability by utilizing structural properties of the success relation between algorithms and test instances. Our findings are significant as they reveal that in spite of the intractability of the problem in its general form, one can indeed exploit sparseness or density of the success relation to obtain non-trivial runtime guarantees for finding an optimal cascading schedule.

## 1 Introduction

When dealing with hard computational problems, one often has access to a *portfolio* of different algorithms that can be applied to solve the given problem, with each of the algorithms having complementary strengths. There are various ways of how this performance complementarity can be exploited. Algorithm selection, a line of research initiated by Rice [19], studies various approaches one can use to select algorithms from the portfolio. Algorithm selection has proven to be an extremely powerful tool with many success stories in Propositional Satisfiability, Constraint Satisfaction, Planning, QBF Solving, Machine Learning and other domains [12, 13, 14, 20]. A common approach to algorithm selection is *per-instance-based algorithm selection*, where an algorithm is chosen for each instance independently, based on some features of the instance (see, *e.g.*, [15, 10]). However, sometimes information about the individual instances is not available or difficult to use. Then, one can instead make use of information about the distribution of the set of instances, *e.g.*, in terms of a representative sample of instances which can be used as a *training set*. In such cases, one can compute in an offline phase a suitable linear ordering of the algorithms, optimizing the ordering for the training set of instances. This ordering is then applied uniformly to any given problem instance in an online fashion—in particular, if the first algorithm in our ordering fails to solve a given instance (due to timeout, memory overflow, or due to not reaching a desired accuracy), then the second algorithm is called, and this continues until we solve the instance. Such a static algorithm selection, "*cascading portfolio scheduling*", is simpler to implement than per-instance selection methods and can be very effective [22]. One prominent recent application of cascading portfolio scheduling lies in state-of-the-art *ImageNet classification models*, where it resulted in a significant speedup by reducing the number of floating-point operations [23]. Cascading portfolio scheduling is also related to online portfolio scheduling [11, 16].

In this paper we address the fundamental problem of finding an optimal cascading schedule for a given portfolio $\mathcal{A}$ of algorithms with respect to a given training set $T$ of instances. In particular, for the problem CASCADING PORTFOLIO SCHEDULING (or CPS for short) that we consider, we are given $m$ algorithms, $n$ test instances, and a *cost mapping* cost, where $\text{cost}(\alpha, t)$ denotes the cost of running algorithm $\alpha$ on test instance $t$, and a *success relation* $S$ where $(\alpha, t) \in S$ means that algorithm $\alpha$ succeeds on test instance $t$. As the cost mapping and the success relation are defined independently, this setting is very general and entails different scenarios.

**Scenario 1** Each algorithm is run until a globally set timeout $C$ is reached. If the algorithm $\alpha$ solves test instance $t$ in time $c \leq C$ then $\text{cost}(\alpha, t) = c$ and $(\alpha, t) \in S$; otherwise we have $\text{cost}(\alpha, t) = C$ and $(\alpha, t) \notin S$.

**Scenario 2** Algorithm $\alpha$ solves a test instance $t$ in time $c$ and outputs an accuracy estimate $r$ for its solution. $r$ is then compared with a globally set accuracy threshold $R$. If $r \geq R$ then $(\alpha, t) \in S$, otherwise $(\alpha, t) \notin S$; in any case $\text{cost}(\alpha, t) = c$. Such a strategy has been used for prediction model generation [23].

**Scenario 3** All the algorithms are first run with a short timeout and if the test instance has not been solved after this, algorithms are run again without a timeout (a similar strategy has been used for QBF solving [18]). Such a strategy can be instantiated to our setting by adding two copies of each algorithm to the portfolio, one with a short timeout and one without a timeout.

**Contribution.** We establish the *fixed-parameter tractability*[1] of computing an optimal cascading schedule by utilizing structural properties of the success relation. We look at the success relation in terms of a Boolean matrix, the *evaluation matrix*, where each row corresponds to a test instance and each column corresponds to an algorithm. A cell contains the entry 1 iff the corresponding algorithm succeeds on the corresponding test. We show that if this matrix is either very sparse or very dense, then the computation of an optimal schedule is tractable. More specifically, we establish the following results, which we describe by writing CPS[*parm*] for CASCADING PORTFOLIO SCHEDULING parameterized by parameter *parm*.

First we consider the *algorithm failure degree* which is the largest number of tests a single algorithm fails on, and the *test failure degree* which is the largest number of algorithms that fail on a single test (these two parameters can also be seen as the largest number of 0's that appear in a row and the largest number of 0's that appear in a column of the matrix, respectively).

 (1) CPS[*algorithm failure degree*] and CPS[*test failure degree*] are fixed-parameter tractable (Theorems 4 and 5).

It is natural to consider also the dual parameters *algorithm success degree* and *test success degree*. However, it follows from known results that CPS is already NP-hard if both of these parameters are bounded by a constant (Proposition 6). Hence, our results exhibit a certain asymmetry between failure and success degrees.

We then consider more sophisticated parameters that capture the sparsity or density of the evaluation matrix. The *failure cover number* is the smallest number of rows and columns in the evaluation matrix needed to cover all the 0's in the matrix; similarly, the *success cover number* is the smallest number of rows and columns needed to cover all the 1's. In fact, both parameters can be computed in polynomial time using bipartite vertex cover algorithms [7].

 (2) CPS[*failure cover number*] and CPS[*success cover number*] are fixed-parameter tractable (Corollary 8 and Theorem 16).

These results are significant as they indicate that CASCADING PORTFOLIO SCHEDULING can be solved efficiently as long as the evaluation matrix is sufficiently sparse or dense. Our result for CPS[*failure cover number*] in fact also shows fixed-parameter tractability of the problem for an even more general parameter than success cover number: the treewidth [21] of the bipartite graph between the algorithms and tests, where edges join success pairs. This is our most technical contribution and reveals how a fundamental graphical parameter [see, *e.g.*, 8] can be utilized for algorithm scheduling.

Another natural variant of the problem, $\text{CPS}^{\text{opt}}[length]$, arises by adding an upper bound $\ell$ on the length, *i.e.*, cardinality, of the computed schedule, and asking for a schedule of length $\leq \ell$ of minimum cost. We obtain a complexity classification of the problem under this parameterization as well.

(3) CPS[*length*] can be solved in polynomial time for each fixed bound $\ell$, but is not fixed-parameter tractable parameterized by $\ell$ subject to established complexity assumptions.

An overview of our results is provided in Table 1.

| Parameter | Complexity | Reference |
|---|---|---|
| Algorithm failure degree | FPT | Proposition 4 |
| Test failure degree | FPT | Proposition 5 |
| Algorithm and test success degree | NP-hard (for constant parameters) | Proposition 6 |
| Failure cover number and failure treewidth | FPT | Theorem 7 |
| Success cover number | FPT | Theorem 16 |
| Length | in XP and W[2]-hard | Proposition 3 |

Table 1: An overview of the complexity results presented in this paper.

## 2 Preliminaries

**Problem Definition.** An instance of the CASCADING PORTFOLIO SCHEDULING problem is a tuple $(\mathcal{A}, T, \text{cost}, S)$ comprising:

- a set $\mathcal{A}$ of $m$ algorithms,
- a set $T$ of $n$ tests,
- a cost mapping cost : $(\mathcal{A} \times T) \rightarrow \mathbb{N}$, and
- a success relation $S \subseteq \mathcal{A} \times T$.

Let $\tau$ be a totally ordered subset of $\mathcal{A}$; we call such a set a *schedule*. The *length* of a schedule is its cardinality. We say that $\tau$ is *valid* if for each test $t$ there exists an algorithm $\alpha \in \tau$ such that $(\alpha, t) \in S$. Throughout the paper, we will assume that there exists a valid schedule for our considered instances—or, equivalently, that each test is solved by at least one algorithm.

The *processing cost* of a test $t$ for a valid schedule $\tau = (\alpha_1, \ldots, \alpha_q)$ is defined as $\sum_{i=1}^{j} \text{cost}(\alpha_i, t)$, where $j$ is the *first* algorithm in $\tau$ such that $(\alpha_j, t) \in S$. The *cost* of a valid schedule $\tau$, denoted $\text{cost}(\tau)$, is the sum of the processing costs of all tests in $T$ for $\tau$. The aim in CASCADING PORTFOLIO SCHEDULING is to find a valid schedule $\tau$ of minimum cost.

**Parameterized Complexity.** In parameterized algorithmics [6, 4, 3, 9] the complexity of a problem is studied not only with respect to the input size $n$ but also a parameter $k \in \mathbb{N}$. The most favorable complexity class in this setting is FPT (*fixed-parameter tractable*) which contains all problems that can be solved by an algorithm running in time $f(k) \cdot n^{\mathcal{O}(1)}$, where $f$ is a computable function. Algorithms running in this time are called *fixed-parameter algorithms*. We will also make use of the complexity classes W[2] and XP, where W[2] $\subseteq$ XP. Problems complete for W[2] are are widely believed to not be in FPT. The class XP contains problems that are solvable in time $\mathcal{O}(n^{f(k)})$, where $f$ is a computable function; in other words, problems in XP are polynomial-time solvable when the parameter is bounded by a constant. To obtain our lower bound results, we will need the notion of a parameterized reduction, referred to as FPT-*reduction*, which is in many ways analogous to the standard polynomial-time reductions; the distinction is that a parameterized reduction runs in time $f(k) \cdot n^{\mathcal{O}(1)}$ for some computable function $f$, and provides upper bounds on the parameter size in the resulting instance [4, 3, 6, 17].

We write $\mathcal{O}^*(f(k))$ to denote a function of the form $f(k) \cdot n^{\mathcal{O}(1)}$, where $n$ is the input length and $k$ is the parameter.

**Problem Parameters.** CASCADING PORTFOLIO SCHEDULING is known to be NP-hard [23], and our aim in this paper will be to circumvent this by obtaining parameters that exploit the fine-grained structure in relevant problem instances. We note that we explicitly aim for results which allow for arbitrary cost mappings, since these are expected to consist of large (and often disorderly) numbers in real-life settings. Instead, we will consider parameters that restrict structural properties of the "binary" success relation. To visualize this success relation, it will be useful to view an instance $\mathcal{I}$ as an $m \times n$ matrix $M_{\mathcal{I}}$ where $M_{\mathcal{I}}[i, j] = 1$ if $(\alpha_i, t_j) \in S$ (*i.e.* if the $j$-th test succeeds on the $i$-th algorithm, for some fixed ordering of algorithms and tests), and $M_{\mathcal{I}}[i, j] = 0$ otherwise.

$$
\begin{array}{c}
\begin{array}{ccccc} t_1 & t_2 & t_3 & t_4 & t_5 \end{array} \\
\begin{array}{c} \alpha_1 \\ \alpha_2 \\ \alpha_3 \\ \alpha_4 \end{array}
\begin{pmatrix} 1 & 1 & 1 & 0 & 1 \\ 0 & 0 & 1 & 0 & 1 \\ 0 & 1 & 0 & 1 & 0 \\ 1 & 1 & 1 & 0 & 1 \end{pmatrix}
\qquad
\begin{pmatrix} 1 & 5 & 2 & 7 & 3 \\ 7 & 7 & 3 & 7 & 5 \\ 7 & 1 & 7 & 6 & 7 \\ 2 & 5 & 3 & 7 & 4 \end{pmatrix} \\
\qquad M_\mathcal{I} \qquad\qquad\qquad\qquad C_\mathcal{I}
\end{array}
$$

Figure 1: An instance with 4 algorithms and 5 tests in the setting where (exact) algorithms are executed with a global timeout of 7, as discussed in Scenario 1. On the left is the matrix $M_\mathcal{I}$ representing the success relation. The failure covering number is 3, as witnessed by the highlighted two rows and one column. The matrix $C_\mathcal{I}$ on the right represents the cost relation, with $C_\mathcal{I}[i,j] = \text{cost}[\alpha_i, t_j]$. The instance $\mathcal{I}$ depicted here has a single solution, notably $(\alpha_1, \alpha_3)$.

The two most natural parameters to consider are $m$ and $n$, and these correspond to the number of rows and columns in $M_\mathcal{I}$, respectively. Unfortunately, these two parameters are also fairly restrictive—it is unlikely that instances of interest will have a very small number of algorithms or test instances. Another option would be to use the maximum number of times an algorithm (or test) can fail (or succeed) as a parameter. In particular, the *algorithm success (or failure) degree* is the maximum number of 1's (or 0's, respectively) occurring in any row in $M_\mathcal{I}$. Similarly, we let the *test success (or failure) degree* be the maximum number of 1's (or 0's, respectively) occurring in any column in $M_\mathcal{I}$. Instances where these parameters are small correspond to cases where "almost everything" either fails or succeeds.

A more advanced parameter that can be extracted from $M_\mathcal{I}$ is the covering number, which intuitively captures the minimum number of rows and columns that are needed to "cover" all successes (or failures) in the matrix. More formally, we say that an entry $M_\mathcal{I}[i,j]$ is covered by row $i$ and by column $j$. Then the *success (or failure) covering number* is the minimum value of $r + c$ such that there exist $r$ rows and $c$ columns in $M_\mathcal{I}$ with the property that each occurrence of 1 (or 0, respectively) in $M_\mathcal{I}$ is covered by one of these rows or columns. Intuitively, an instance has success covering number $s$ if there exist $r$ algorithms and $s - r$ tests such that these have a non-empty intersection with every relation in $S$—see Figure 1 for an example. We note that the covering number has been used as a structural parameter of matrices, notably in previous work on the MATRIX COMPLETION problem [7], and that it is possible to compute $r$ algorithms and $c$ tests achieving a minimum covering number in polynomial time [7, Proposition 1]. We will denote the success covering number by $\text{cov}_s$ and the failure covering number by $\text{cov}_f$.

## 3 Results for Basic Parameters

In this section we consider the CASCADING PORTFOLIO SCHEDULING problem parameterized by the *number of algorithms* (*i.e.*, by $m = |\mathcal{A}|$), by the number of tests (*i.e.*, by $n = |T|$), and by the *length* of the computed schedule.

We begin mapping the complexity of our problem with two initial propositions. Note that both propositions can also be obtained as corollaries of the more general Theorem 16, presented later. Still, we consider it useful to present a short sketch of proof of Proposition 1, since it nicely introduces the combinatorial techniques that will later be extended in the proof of Theorem 1.

**Proposition 1.** CPS[*number of algorithms*] is in FPT.

*Proof Sketch.* We reduce the problem to that of finding a minimum-weight path in a directed acyclic graph (DAG) $D$. We construct $D$ as follows. We create a single source vertex $s$, and a single destination vertex $z$ in $D$. We define $L_0 = \{s\}$, $L_{m+1} = \{z\}$, and apart from $z$, $D$ contains $m$ layers, $L_0, \ldots, L_m$, of vertices, where layer $L_i$, for $i \in \{0, \ldots, m\}$, contains a vertex for each subset of $\mathcal{A}$ of cardinality $i$, with vertex $s$ corresponding to the empty set. We connect each vertex that corresponds to a subset of $\mathcal{A}$ which is a valid portfolio to $z$. For each vertex $u$ in layer $L_i$, $i \in \{0, \ldots, m-1\}$, corresponding to a subset $S_u \subset \mathcal{A}$, and each vertex $v \in L_{i+1}$ corresponding to a subset $S_v \subseteq \mathcal{A}$, where $S_v = S_u \cup \{\alpha\}$, for $\alpha \in \mathcal{A}$, we add an edge $(u, v)$ if there exists a test $t \in T$ such that (1) $(\alpha, t) \in S$ and (2) there does not exist $\beta \in S_u$ such that $(\beta, t) \in S$; in such case the weight of $(u, v)$, $wt(u, v)$, is defined as follows. Let $T_\alpha \subseteq T$ be the set of tests that cannot be solved by any algorithm in $S_u$. Then $wt(u, v) = \sum_{t \in T_\alpha} cost(\alpha, t)$. Informally speaking, the weight of $(u, v)$ is the additional cost incurred by appending algorithm $\alpha$ to any (partial) portfolio consisting of the algorithms in $S_u$. This completes the construction of $D$.

It is not difficult to show that an optimal portfolio for $\mathcal{A}$ corresponds to a minimum-weight path from $s$ to $z$, which can be computed in time $\mathcal{O}^*(2^m)$.  □

**Proposition 2.** CPS[*number of tests*] is in FPT.

To formally capture the parameterization of the problem by the length $\ell$ of the computed schedule, we need to slightly adjust its formal definition. Let CPS$^{\text{val}}$[*length*] and CPS$^{\text{opt}}$[*length*] denote the variants of CASCADING PORTFOLIO SCHEDULING where for each problem instance we are also given an integer $\ell > 0$ and only schedules up to length $\ell$ are considered ($\ell$ being the parameter). CPS$^{\text{val}}$[*length*] is the decision problem that asks whether there exists a valid schedule of length $\leq \ell$, and CPS$^{\text{opt}}$[*length*] asks to compute a valid schedule of length $\leq \ell$ of smallest cost or decide that no valid schedule of length $\leq \ell$ exists. Both problems are parameterized by the length $\ell$.

**Proposition 3.** CPS$^{\text{opt}}$[*length*] is in XP, but is unlikely to be in FPT since already CPS$^{\text{val}}$[*length*] is W[2]-complete.

*Proof Sketch.* Membership of CPS$^{\text{opt}}$[*length*] in XP is easy: We enumerate every ordered selection of at most $\ell$ algorithms from $\mathcal{A}$ (there are at most $\mathcal{O}(\ell! m^\ell)$ many) and if valid, we compute its cost, and keep track of a valid selection (if any) of minimum cost over all enumerations.

To prove the W[2]-hardness of CPS$^{\text{val}}$[*length*], we give an FPT-reduction from the W[2]-complete problem SET COVER [4]. The membership of CPS$^{\text{val}}$[*length*] in W[2] follows from a straightforward reduction to SET COVER, which is omitted.

Given an instance $((U, \mathcal{F}), k)$ of SET COVER, where $U$ is a ground set of elements, $\mathcal{F}$ is a family of subsets for $U$, and $k \in \mathbb{N}$ is the parameter, we create an instance of CASCADING PORTFOLIO SCHEDULING as follows. We set $T = U$, and for each $F \in \mathcal{F}$, we create an algorithm $\alpha_F \in \mathcal{A}$ and add $(\alpha_F, t)$ to $S$, for every $t \in F$. Finally, we set $\ell = k$. The function *cost* can be defined arbitrarily. The above reduction is clearly a (polynomial-time) FPT-reduction, and it is straightforward to verify that $((U, \mathcal{F}), k)$ is a yes-instance of SET COVER if and only if the constructed instance of CASCADING PORTFOLIO SCHEDULING has a valid portfolio of size at most $\ell$.  □

We remark that the above construction can also be used to show that the problem variants arising in Scenarios 1-3 described in the introduction remain W[2]-complete.

## 4 Results for Degree Parameters

This section presents a classification of the complexity of CASCADING PORTFOLIO SCHEDULING parameterized by the considered (success and failure) degree parameters.

**Proposition 4.** CPS[*algorithm failure degree*] is in FPT.

*Proof.* Denote by $\deg_f^{\mathcal{A}}$ the algorithm failure degree, and let $\mathcal{I} = (\mathcal{A}, T, \text{cost}, S)$ be an instance of CASCADING PORTFOLIO SCHEDULING. Consider an algorithm which loops over each algorithm $\alpha \in \mathcal{A}$ and proceeds under the assumption that $\alpha$ is the first algorithm in an optimal valid portfolio. For each such $\alpha$, the number of tests in $T$ that cannot be evaluated by $\alpha$ is at most $\deg_f^{\mathcal{A}}$. Removing $\alpha$ from $\mathcal{A}$ and the subset of tests $\{t \mid (\alpha, t) \in S\}$ from $T$ results in an instance $\mathcal{I}^-$ of CASCADING PORTFOLIO SCHEDULING with at most $\deg_f^{\mathcal{A}}$ tests, which, by Proposition 2, can be solved in time $\mathcal{O}^*((\deg_f^{\mathcal{A}})^{\deg_f^{\mathcal{A}}})$ to obtain an optimal solution for $\mathcal{I}^-$. Prefixing $\alpha$ to the optimal solution obtained for $\mathcal{I}^-$ (assuming a solution exists) results in an optimal solution $S_\alpha$ for $\mathcal{I}$ under the constraint that algorithm $\alpha$ is the first algorithm. Enumeration every algorithm $\alpha \in \mathcal{A}$ as the first algorithm, computing $S_\alpha$, and keeping track of the solution of minimum cost over all enumerations, results in an optimal solution for $\mathcal{I}$. The running time of the above algorithm is $\mathcal{O}^*((\deg_f^{\mathcal{A}})^{\deg_f^{\mathcal{A}}})$.  □

**Proposition 5.** CPS[*test failure degree*] is in FPT.

*Proof.* Denote by $\deg_f^T$ the test failure degree, and let $\mathcal{I} = (\mathcal{A}, T, \text{cost}, S)$ be an instance of CASCADING PORTFOLIO SCHEDULING. Consider an algorithm which (1) loops over each algorithm $\alpha \in \mathcal{A}$ and proceeds under the assumption that $\alpha$ is the last algorithm in an optimal valid portfolio $\tau$, and then (2) loops over every test $t$ in our instance and proceed under the assumption that $t$ is a test that is solved *only* by $\alpha$ in $\tau$. For each such choice of $t$ and $\alpha$, it follows that the algorithms preceding $\alpha$ in $\tau$ *do not* solve $t$, and hence there are at most $\deg_f^T$ many such algorithms. Therefore, we can check the validity and compute the cost of every possible ordered selection of a subset from these algorithms that precede $\alpha$ in $\tau$. After we finish looping over all choices of $\alpha$ and $t$, we output a valid portfolio of minimum cost.

There are $|\mathcal{A}|$ choices for a last algorithm $\alpha$ and $|T|$ choices for a desired test $t$. For each fixed $\alpha$ and $t$, there are at most $\mathcal{O}^*((\deg_f^T)!)$ many ordered selections of a subset of algorithms preceding $\alpha$ in $\tau$. It follows that the problem can be solved in time $\mathcal{O}^*((\deg_f^T)!)$. $\qquad\square$

**Proposition 6.** CPS[*algorithm success degree*], CPS[*test success degree*], and even CPS[*algorithm success degree + test success degree*] are NP-hard already if the algorithm success degree is at most 3 and test success degree is at most 2.

*Proof.* We reduce from the problem 3-MIN SUM VERTEX COVER, where we are given a graph $H = (V, E)$ with maximum degree 3, and the task is to find a bijection $\sigma : V \rightarrow \{1, \ldots, V\}$ that minimizes $\sum_{e \in E} f_\sigma(e)$, where $f_\sigma(e) = \min_{v \in e} \sigma(v)$. Feige *et al.* [5] showed that there exists $\epsilon > 0$ such that it is NP-hard to approximate 3-MIN SUM VERTEX COVER within a ratio better than $1 + \epsilon$. Given an instance of this problem, we construct an instance of $(\mathcal{A}, T, \text{cost}, S)$ of CASCADING PORTFOLIO SCHEDULING by letting $\mathcal{A} = V$, adding for each edge $e \in E$ a test $t_e$ to $T$, setting $S = \{(\alpha, t_e) \in \mathcal{A} \times T : \alpha \in e\}$, and setting $\text{cost}(\alpha, t) = 1$ for all $\alpha \in \mathcal{A}$ and $t \in \mathcal{T}$. It is easy to verify that bijections $\sigma$ that minimize $\sum_{e \in E} f_\sigma(e)$ are exactly those that give an ordering $\tau$ of $\mathcal{A}$ of minimal cost. It remains to observe that the the algorithm success degree is 3 and the test success degree is 2. $\qquad\square$

# 5 Results for Cover Numbers

In this section we show that CPS[*failure cover number*] and CPS[*success cover number*] are both fixed-parameter tractable.

## 5.1 Using the Failure Cover Number

The first of the two results follows from an even more general result, the fixed-parameter tractability of CPS[*failure treewidth*], where as the parameter we take the *treewidth* of the failure graph $G_\mathcal{I}$ defined as follows.

The failure graph $G_\mathcal{I}$ is a bipartite graph whose vertices consist of $\mathcal{A} \cup T$ and where there is an edge between $\alpha \in \mathcal{A}$ and $t \in T$ iff $t$ fails on $\mathcal{A}$. We note that the algorithm (or test) failure degree naturally corresponds to the maximum degree in the respective bipartition of $G_\mathcal{I}$, and that the failure covering number is actually the size of a minimum vertex cover in $G_\mathcal{I}$.

Treewidth [21, 8, 1] is a well-established graph parameter that measures the "tree-likeness" of instances. Aside from treewidth, we will also need the notion of *balanced separators* in graphs. We introduce these technical notions below.

**Treewidth and Separators.** Let $G = (V, E)$ be a graph. A *tree decomposition* of $G$ is a pair $(\mathcal{V}, \mathcal{T})$ where $\mathcal{V}$ is a collection of subsets of $V$ such that $\bigcup_{X_i \in \mathcal{V}} = V$, and $\mathcal{T}$ is a rooted tree whose node set is $\mathcal{V}$, such that:

1. For every edge $\{u, v\} \in E$, there is an $X_i \in \mathcal{V}$, such that $\{u, v\} \subseteq X_i$; and
2. for all $X_i, X_j, X_k \in \mathcal{V}$, if the node $X_j$ lies on the path between the nodes $X_i$ and $X_k$ in the tree $\mathcal{T}$, then $X_i \cap X_k \subseteq X_j$.

The *width* of the tree decomposition $(\mathcal{V}, \mathcal{T})$ is defined to be $\max\{|X_i| \mid X_i \in \mathcal{V}\} - 1$. The *treewidth* of the graph $G$, denoted $\text{tw}(G)$, is the minimum width over all tree decompositions of $G$.

A pair of vertex subsets $(A, B)$ is a *separation* in graph $G$ if $A \cup B = V(G)$ and there is no edge between $A \setminus B$ and $B \setminus A$. The *separator* of this separation is $A \cap B$, and the *order* of separation $(A, B)$ is equal to $|A \cap B|$. We say that a separation $(A, B)$ of G is an $\alpha$-*balanced* separation if $|A \setminus B| \leq \alpha |V(G)|$ and $|B \setminus A| \leq \alpha |V(G)|$.

**Proof Strategy.** Our main aim in this section will be to prove the following theorem:

**Theorem 7.** CPS[*failure treewidth*] is in FPT.

It is easy to see that failure treewidth is at most the failure cover number plus 1 (consider, *e.g.*, a tree decomposition of the failure graph consisting of a sequence of bags, each containing the algorithms and tests forming the cover and one additional test or algorithm). Hence, once we establish Theorem 7 we obtain the following as an immediate corollary:

**Corollary 8.** CPS[*failure cover number*] is in FPT.

We first provide below a high-level overview of the proof of Theorem 7.

We solve the problem using dynamic programming on a tree decomposition of $G_{\mathcal{I}}$, by utilizing the upper bound on the solution length derived in the first step. The running time is $\mathcal{O}^*(4^{\mathrm{tw}(G_{\mathcal{I}})} \cdot \mathrm{tw}(G_{\mathcal{I}})^{\mathrm{tw}(G_{\mathcal{I}})})$. To make the dynamic programming approach work, for a current bag in the tree decomposition, and for each test in the bag, we remember whether the test is solved by an algorithm in the future or by an algorithm in the past. Moreover, we remember which tests are solved by the same algorithm. We also remember specifically which algorithm is the "first" from the future and which is the "first" from the past. Finally, we remember the relative positions of the algorithms in the bag, the first algorithm from the future, the first algorithm from the past, and the algorithms that solve the tests in the bag. Note that we do not remember which algorithms solve tests in the bag, only their relative position and whether they are in the past or future.

We now turn to giving a more detailed proof for Theorem 7.

**Lemma 9.** *A minimum cost schedule for* CASCADING PORTFOLIO SCHEDULING *can be computed in time* $\mathcal{O}^*(4^{\mathrm{tw}} \cdot \mathrm{tw}^{\mathrm{tw}})$.

*Proof Sketch.* As with virtually all fixed-parameter algorithms parameterized by treewidth, we use leaf-to-root dynamic programming along a tree decomposition (in this case of the failure graph $G_{\mathcal{I}}$)—see for instance the numerous examples presented in the literature [4, 3]. However, due to the specific nature of our problem, the records dynamically computed by the program are far from standard. This can already be seen by considering the size of our records: while most such dynamic programming algorithms only store records that have size bounded by a function of the treewidth, in our case the records will also have a polynomial dependence on $m$.

As a starting point, we will use the known algorithm of Bodlaender *et al.* [2] to compute a tree-decomposition of width at most $5 \cdot \mathrm{tw}(G_{\mathcal{I}})$. We proceed by formalizing the used records. Let $X_i$ be a bag in the tree decomposition. A *configuration* w.r.t. $X_i$ is a tuple $(\alpha_{\mathrm{past}}, \alpha_{\mathrm{future}}, \sigma, \delta)$, where

- $\alpha_{\mathrm{past}}$ is an algorithm that has been forgotten in a descendant of $X_i$,
- $\alpha_{\mathrm{future}}$ is an algorithm that has not been introduced yet in $X_i$,
- $\sigma : X_i \cup \{\alpha_{\mathrm{past}}, \alpha_{\mathrm{future}}\} \to [|X_i| + 2]$, and
- $\delta : T \cap X_i \to \{\text{"past"}, \text{"future"}\}$.

Note that there are at most $2^{|X_i|} \cdot (|X_i| + 2)^{|X_i|+2} \cdot m^2 = \mathcal{O}^*(2^{\mathrm{tw}} \cdot \mathrm{tw}^{\mathrm{tw}})$ configurations. The interpretation of the configuration is that $\sigma$ tells us the relative positions in the final schedule of the algorithms in $X_i$, $\alpha_{\mathrm{past}}$, $\alpha_{\mathrm{future}}$, and for each test in $X_i$ the algorithm that finally solves the test $t$. The function $\delta$, for a test $t$, tells us whether the algorithm that is the first in the schedule that solves $t$ was already introduced ("past") or will be introduced ("future"). The entry $\alpha_{\mathrm{past}}$ represents the specific algorithm that is first in the schedule among all algorithms that have been already forgotten in the descendant, and $\alpha_{\mathrm{future}}$ that among the ones that have not been introduced yet.

We say that a configuration $C = (\alpha_{\mathrm{past}}, \alpha_{\mathrm{future}}, \sigma, \delta)$ w.r.t. $X_i$ is *admissible* if

- for all algorithms $\alpha_1, \alpha_2 \in \mathcal{A} \cap (X_i \cup \{\alpha_{\mathrm{past}}, \alpha_{\mathrm{future}}\})$, it holds that $\sigma(\alpha_1) \neq \sigma(\alpha_2)$;
- for all $t \in T \cap X_i$ if $\sigma(t) = j$, then for every $j' < j$: if there is $\alpha \in \mathcal{A} \cap (X_i \cup \{\alpha_{\mathrm{past}}, \alpha_{\mathrm{future}}\})$ such that $\sigma(\alpha) = j'$ then $\alpha$ does not solve $t$;
- for all $t \in T \cap X_i$ if $\delta(t) = \text{"past"}$, then either $\sigma(\alpha_{\mathrm{past}}) \leq \sigma(t)$ or there is $\alpha \in \mathcal{A} \cap X_i$ such that $\sigma(\alpha) = \sigma(t)$;
- for all $t \in T \cap X_i$ if $\delta(t) = \text{"future"}$, then $\sigma(\alpha_{\mathrm{future}}) \leq \sigma(t)$;
- for all $j', j \in [|X_i| + 2]$ such that $j' < j$, if $\sigma^{-1}(j') = \emptyset$, then $\sigma^{-1}(j) = \emptyset$; and
- if $\sigma(\alpha) = \sigma(t)$ for some $\alpha \in \mathcal{A} \cap (X_i \cup \{\alpha_{\mathrm{past}}\})$ and $t \in T \cap X_i$, then $\delta(t) = \text{"past"}$ and $\alpha$ solve $t$.

Note that if we take any valid schedule, we can project it w.r.t. a bag $X_i$ and obtain a configuration $(\alpha_{\mathrm{past}}, \alpha_{\mathrm{future}}, \sigma, \delta)$. Such a configuration will always be admissible and so we can restrict our attention to admissible configurations only. To simplify the notation we let $\Gamma_i[C] = \infty$ if $C$ is not an admissible configuration w.r.t. $X_i$.

Now for each $X_i$, we will compute a table $\Gamma_i$ that contains an entry for each admissible configuration $C$ such that $\Gamma_i[C] \in \mathbb{N}$ is the best cost, w.r.t. configuration $C$, of the already introduced tests restricted to the already introduced algorithms and the algorithm $\alpha_{\mathrm{future}}$.

Clearly, the minimum cost schedule of the instance gives rise to some admissible configuration $C$ w.r.t. the root node $X_r$ of the tree decomposition. Hence $\Gamma_r[C]$ contains the minimum cost of a schedule. To complete the proof, it suffices to show how to update the records when traversing the tree-decomposition in dynamic fashion. Below, we list the sequence of claims (along with some exemplary proofs) used to this end.

**Claim 10.** *If $X_i$ is a leaf node, then $\Gamma_i$ can be computed in $\mathcal{O}(|\Gamma_i|)$ time.*

*Proof of Claim.* Note that $X_i = \emptyset$ and that none of the algorithms has been introduced in any leaf node. The only admissible configuration looks like $(\emptyset, \alpha, \{(\alpha, 0)\}, \emptyset)$, where $\alpha \in \mathcal{A}$. Moreover, since no tests or algorithms were introduced at that point, the cost of all of these configurations is zero. $\qquad\square$

**Claim 11.** *If $X_i$ is an introduce node for a test with the only child $X_j$, then $\Gamma_i$ can be computed in $\mathcal{O}(|\Gamma_i|)$ time.*

**Claim 12.** *If $X_i$ is an introduce node for an algorithm with the only child $X_j$, then $\Gamma_i$ can be computed in $\mathcal{O}(|\Gamma_i|)$ time.*

**Claim 13.** *If $X_i$ is a forget node, which forgets a test $t$, with the only child $X_j$, then $\Gamma_i$ can be computed in $\mathcal{O}(\ell|\Gamma_i|)$ time.*

*Proof of Claim.* Let $C = (\alpha_{\text{past}}, \alpha_{\text{future}}, \sigma, \delta)$ be an admissible configuration w.r.t. $X_i$. Forgetting a test does not change the costs of introduced tests w.r.t. introduced algorithms. Hence, we only need to find a configuration w.r.t. $X_j$ of the lowest cost that after removing $t$ from $\delta$ results in $C$. Let $\delta_p$ be a function we get from $\delta$ by adding $\delta_p(t) = $ "past" and let $\delta_f$ be a function we get from $\delta$ by adding $\delta_f(t) = $ "future". First let $C_f$ be a configuration $(\alpha_{\text{past}}, \alpha_{\text{future}}, \sigma_f, \delta_f)$ such that $\sigma_f(x) = \sigma(x)$ for all $x \in (X_i \cup \{\alpha_{\text{past}}, \alpha_{\text{future}}\}) \setminus \{t\}$ and $\sigma_f(t) = \sigma(\alpha_{\text{future}})$. Now, for $k \in [|X_i| + 2]$ and let $C_k^1$ be a configuration $(\alpha_{\text{past}}, \alpha_{\text{future}}, \sigma_k^1, \delta_p)$ such that $\sigma_k^1(x) = \sigma(x)$ for all $x \in (X_i \cup \{\alpha_{\text{past}}, \alpha_{\text{future}}\}) \setminus \{t\}$ and $\sigma_k^1(t) = k$ and let $C_k^2$ be a configuration $(\alpha_{\text{past}}, \alpha_{\text{future}}, \sigma_k^2, \delta_p)$ such that $\sigma_k^2(x) = \sigma(x)$ for all $x \in (X_i \cup \{\alpha_{\text{past}}, \alpha_{\text{future}}\}) \setminus \{t\}$ such that $\sigma(x) < k$, $\sigma_k^2(x) = \sigma(x) + 1$ for all $x \in (X_i \cup \{\alpha_{\text{past}}, \alpha_{\text{future}}\}) \setminus \{t\}$ such that $\sigma(x) \geq k$, and $\sigma_k^1(t) = k$. Note that $\sigma_k^2$ would be also shifted to $\sigma$ after removing the entry for $t$.

We let $\Gamma_i[C]$ be minimum among $C_f$ and $\min_{k \in [|X_i|+2], \ell \in \{1,2\}} \Gamma_j[C_k^\ell]$. $\qquad\square$

**Claim 14.** *If $X_i$ is a forget node, which forgets an algorithm $\alpha$, with the only child $X_j$, then $\Gamma_i$ can be computed in $\mathcal{O}((\ell + m)|\Gamma_i|)$ time.*

*Proof of Claim.* Let $C = (\alpha_{\text{past}}, \alpha_{\text{future}}, \sigma, \delta)$ be an admissible configuration w.r.t. $X_i$. Clearly, when we forget an algorithm, the cost of schedule given by $\sigma$ w.r.t. already introduced algorithms and tests does not change. Hence, we just need to choose the best configuration of $X_j$ that can result in $C$.

We distinguish two cases depending on whether $\alpha_{\text{past}} = \alpha$ or not.

First, if $\alpha_{\text{past}} = \alpha$, then for an already forgotten algorithm $\alpha'$, $k \in [|X_i| + 2]$ such that $\sigma(\alpha_{\text{past}}) \geq k$, and $\ell \in \{0, 1\}$ let us denote by $C_{\alpha', k, \ell}$ the configuration $(\alpha', \alpha_{\text{future}}, \sigma_{\alpha', k}^\ell, \delta)$ such that $\sigma_{\alpha', k}^\ell(\alpha') = k$, for all $x \in X_i \cup \{\alpha_{\text{past}}, \alpha_{\text{future}}\}$ $\sigma_{\alpha', k}^\ell(x) = \sigma(x)$ if $\sigma(x) < k$ and $\sigma_{\alpha', k}^\ell(x) = \sigma(x) + \ell$ otherwise. Note that in order for $\sigma_{\alpha', k}^0$ to be admissible, $\sigma^{-1}(k)$ contains at least one test and no algorithm. In this case we let $\Gamma_i[C] = \min_{\alpha', k, \ell} \Gamma_j[C_{\alpha', k, \ell}]$.

If $\alpha_{\text{past}} \neq \alpha$, then for $k \in [|X_i| + 2]$ such that $\sigma(\alpha_{\text{past}}) < k$, and $\ell \in \{0, 1\}$ let us denote by $C_{k, \ell}$ the configuration $(\alpha_{\text{past}}, \alpha_{\text{future}}, \sigma_k^\ell, \delta)$ such that $\sigma_k^\ell(\alpha) = k$, for all $x \in X_i \cup \{\alpha_{\text{past}}, \alpha_{\text{future}}\}$ $\sigma_k^\ell(x) = \sigma(x)$ if $\sigma(x) < k$ and $\sigma_k^\ell(x) = \sigma(x) + \ell$ otherwise. Note that again in order for $\sigma_k^0$ to be admissible, $\sigma^{-1}(k)$ contains at least one test and no algorithm. In this case we let $\Gamma_i[C] = \min_{k, \ell} \Gamma_j[C_{k, \ell}]$. $\qquad\square$

**Claim 15.** *If $X_i$ is a join node with children $X_{j_1}$ and $X_{j_2}$, then $\Gamma_i$ can be computed from $\Gamma_{j_1}$ and $\Gamma_{j_2}$ in $\mathcal{O}(2^\ell m |\Gamma_i|)$ time.*

To conclude, the last four claims show that it is possible to dynamically compute our records from the leaves of a nice tree decomposition to its root; once the records are known for the root, the algorithm has all the information it needs to output with the solution. $\qquad\square$

It follows that CPS[*failure treewidth*] is fixed-parameter tractable, hence establishing Theorem 7.

## 5.2 Using the Success Cover Number

The aim of this section is to establish the fixed-parameter tractability of CPS[*success cover number*], which can be viewed as a dual result to Corollary 8. The techniques used to obtain this result are entirely different from those used in the previous subsection; in particular, the proof is based on a significant extension of the ideas introduced in the proof of Proposition 1.

**Theorem 16.** CPS[*success cover number*] is in FPT.

*Proof Sketch.* Let $\mathcal{I}$ be an instance of CPS[$\text{cov}_s$]. Our first step is to compute a witness for the success cover number $\text{cov}_s$, *i.e.*, a set of algorithms $\mathcal{A}'$ and tests $T'$ such that $|\mathcal{A}' \cup T'| = \text{cov}_s$ and each pair in $S$ has a non-empty intersection with $\mathcal{A}' \cup T'$; as discussed in Subsection 2, this can be done in polynomial time [7, Proposition 1]. Let $V = 2^{\mathcal{A}' \cup T'}$ be the set of all subsets of $\text{cov}_s$. We will construct a directed arc-weighted graph $D$ with vertex set $V \cup \{x\}$, and with the property that each shortest path from $\emptyset$ to $x$ precisely corresponds to a minimum-cost schedule for the input instance $\mathcal{I}$. Intuitively, reaching a vertex $v$ in $D$ which corresponds to a certain set of algorithms $\mathcal{A}_0$ and tests $T_0$ means that the schedule currently contains the algorithms in $\mathcal{A}_0$ plus an optimal choice of algorithms which can process the remaining tests in $T_0$; information about the ordering inside the schedule is not encoded by the vertex $v$ itself, but rather by the path from $\emptyset$ to $v$.

In order to implement this idea, we will add the following arcs to $D$. To simplify the description, let $\mathcal{A}^*$ be an arbitrary subset of $\mathcal{A}'$ and $T^*$ be an arbitrary subset of $T'$. First of all, for each $\mathcal{A}^*$ such that for every test $t \in T \setminus T'$ there is some $\alpha \in \mathcal{A}^*$ satisfying $(\alpha, t) \in S$, we add the arc $(\mathcal{A}^* \cup T', x)$ and assign it a weight of $0$. This is done to indicate that $\mathcal{A}^* \cup T'$ corresponds to a valid schedule.

Second, for each $\mathcal{A}^*$ that is a proper subset of $\mathcal{A}'$, $\alpha_0 \in \mathcal{A}' \setminus \mathcal{A}^*$, and $T^*$, we add the arc $e$ from $\mathcal{A}^* \cup T^*$ to $\mathcal{A}^* \cup \{\alpha_0\} \cup T^* \cup T_0$, where $T_0$ contains every test $t_0 \in T'$ such that $(\alpha_0, t_0) \in S$. In order to compute the weight of this arc $e$, we first compute the set $T_e$ of all tests outside of $T^*$ where $\alpha_0$ will be queried (assuming $\alpha_0$ is added to the schedule at this point); formally, $t \in T_e$ if $t \notin T^*$ and for each $\alpha' \in \mathcal{A}^*$ it holds that $(\alpha', t) \notin S$. For clarity, observe that $T_0 \subseteq T_e$. Now, we set the weight of $e$ to $\sum_{t \in T_e} \text{cost}(\alpha_0, t)$.

To add our third and final set of edges, we first pre-compute for each $T_\lambda \subseteq T' \setminus T^*$ an algorithm $\alpha_\lambda \in \mathcal{A} \setminus \mathcal{A}'$ such that:

1. for each $t_\lambda \notin T^*$, $(\alpha_\lambda, t_\lambda) \in S$ iff $t_\lambda \in T_\lambda$ (*i.e.*, $\alpha_\lambda$ successfully solves exactly $T_\lambda$), and

2. among all possible algorithms satisfying the above condition, $\alpha_\lambda$ achieves the *minimum cost* for all as-of-yet-unprocessed tests. Formally, $\alpha_\lambda$ minimizes the term $\text{price}(\alpha_\lambda) = \left( \sum_{t \in (T' \setminus T^*)} \text{cost}(\alpha_\lambda, t) \right) + \left( \sum_{t \notin T': \forall \alpha \in \mathcal{A}^*: (\alpha, t) \notin S} \text{cost}(\alpha_\lambda, t) \right)$.

Now, we add an arc $e$ from each $\mathcal{A}^* \cup T^*$ to each $\mathcal{A}^* \cup T^* \cup T_\lambda$, where $T_\lambda$ is defined as above and associated with the test $\alpha_\lambda$. The weight of $e$ is precisely the value $\text{price}(\alpha_\lambda)$.

Note that since the graph $D$ has $2_s^{\text{cov}} + 1$ many vertices, a shortest path $P$ from $\emptyset$ to $x$ in $D$ can be computed in time $2^{\mathcal{O}(\text{cov}_s)}$. Moreover, it is easy to verify that $D$ can be constructed from an instance $\mathcal{I}$ in time at most $2^{\mathcal{O}(\text{cov}_s)} \cdot |\mathcal{I}|^2$. At this point, it remains to verify that a shortest $\emptyset$-$x$ path $P$ in $D$ can be used to obtain a solution for $\mathcal{I}$. $\qquad\square$

## 6 Conclusion

We studied the parameterized complexity of the CASCADING PORTFOLIO SCHEDULING problem under various parameters. We identified several settings where the NP-hardness of the problem can be circumvented via exact fixed-parameter algorithms, including cases where (i) the algorithms have a small failure degree, (ii) the tests have a small failure degree, (iii) the evaluation matrix has a small failure cover, and (iv) the evaluation matrix has a small success cover. The first three cases can be seen as settings in which most algorithms succeed on most of the tests, whereas case (iv) can be seen as a setting where most algorithms fail.

We have complemented our algorithmic results with hardness results which allowed us to draw a detailed complexity landscape of the problem. We would like to point out that all our hardness results hold even when all costs are unit costs. This finding is significant, as it reveals that the complexity of the problem mainly depends on the success relation and not on the cost mapping.

For future work, it would be interesting to extend our study to the more complex setting where up to $p$ algorithms from the portfolio can be run in parallel. Here, the number $p$ could be seen as a natural additional parameter.

**Acknowledgments**

Robert Ganian acknowledges the support by the Austrian Science Fund (FWF), Project P 31336, and is also affiliated with FI MUNI, Brno, Czech Republic. Stefan Szeider acknowledges the support by the Austrian Science Fund (FWF), Project P 32441.

## Footnotes

[1]Fixed-parameter tractability is a relaxation of polynomial tractability; definitions are provided in Section 2.

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
