[Supplementary Material · FullVersion.pdf]

# The Parameterized Complexity of
# Cascading Portfolio Scheduling
# (Full Version)

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

 $t$. More specifically, if $P = (v_0 = s, v_1, \ldots, v_r, t)$ is a minimum-weight $s$-$t$ path in $D$, then the schedule $(\alpha_1, \ldots, \alpha_r)$, where $\{\alpha_i\} = S_{v_i} \setminus S_{v_{i-1}}$ ($S_{v_i}$ is the subset of algorithms corresponding to $v_i$ and $S_{v_{i-1}}$ that corresponding to $v_{i-1}$), for $i \in [r]$, is an optimal schedule for $T$. The number of vertices in $D$ is $2^m + 1$, the number of edges $\mathcal{O}(2^m m^2)$, and computing a minimum-weight path in $D$ can be done in linear time in the size of $D$ [3], which is $\mathcal{O}^*(2^m)$. It follows that the problem can be solved in time $\mathcal{O}^*(2^m)$. $\qquad\square$

**Proposition 2.** CPS[*number of tests*] is in FPT.

To formally capture the parameterization of the problem by the length $\ell$ of the computed schedule, we need to slightly adjust its formal definition. Let $\text{CPS}^{\text{val}}[length]$ and $\text{CPS}^{\text{opt}}[length]$ denote the variants of CASCADING PORTFOLIO SCHEDULING where for each problem instance we are also given an integer $\ell > 0$ and only schedules up to length $\ell$ are considered ($\ell$ being the parameter). $\text{CPS}^{\text{val}}[length]$ is the decision problem that asks whether there exists a valid schedule of length $\leq \ell$,

185   and CPS^opt[*length*] asks to compute a valid schedule of length $\leq \ell$ of smallest cost or decide that no
186   valid schedule of length $\leq \ell$ exists. Both problems are parameterized by the length $\ell$.

**Proposition 3.** CPS^val[*length*] is in XP, but is unlikely to be in FPT since already CPS^val[*length*] is
188   W[2]-complete.

*Proof.* Membership of CPS^opt[*length*] in XP is easy: We enumerate every ordered selection of at
190   most $\ell$ algorithms from $\mathcal{A}$ (there are at most $\mathcal{O}(\ell! m^{\ell})$ many) and if valid, we compute its cost, and
191   keep track of a valid selection (if any) of minimum cost over all enumerations.

To prove the W[2]-completeness of CPS^val[*length*], we give FPT-reductions from and to the W[2]-
193   complete problem SET COVER [5]. The reduction to SET COVER (showing membership in W[2]) is
194   straightforward: The set of tests $T$ is the ground set $U$ of the constructed instance of SET COVER, and
195   for each algorithm $\alpha \in \mathcal{A}$, we add to the family of subsets of $U$, $\mathcal{F}$, a set $F_\alpha = \{t \in T \mid (\alpha, t) \in S\}$.
196   It is clear that the resulting instance of SET COVER has a cover of size at most $\ell$ iff there a valid
197   cascading portfolio scheduling of length at most $\ell$, and hence the above reduction is an FPT-reduction
198   to SET COVER.

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

 tree decomposition $(\mathcal{V}, \mathcal{T})$ is *nice* if it satisfies the following conditions:

1. Each node in the tree $\mathcal{T}$ has at most two children.

2. If a node $X_i$ has two children $X_j$ and $X_k$ in the tree $\mathcal{T}$, then $X_i = X_j = X_k$; in this case node $X_i$ is called a *join node*.

3. If a node $X_i$ has only one child $X_j$ in the tree $\mathcal{T}$, then either $|X_i| = |X_j| + 1$ and $X_j \subset X_i$, and in this case $X_i$ is called an *insert node*; or $|X_i| = |X_j| - 1$ and $X_i \subset X_j$, and in this case $X_i$ is called a *forget node*.

4. If $X_i$ is a leaf node or the root, then $X_i = \emptyset$.

A pair of vertex subsets $(A, B)$ is a *separation* in graph $G$ if $A \cup B = V(G)$ and there is no edge between $A \setminus B$ and $B \setminus A$. The *separator* of this separation is $A \cap B$, and the *order* of separation $(A, B)$ is equal to $|A \cap B|$. We say that a separation $(A, B)$ of G is an $\alpha$-*balanced* separation if $|A \setminus B| \leq \alpha|V(G)|$ and $|B \setminus A| \leq \alpha|V(G)|$.

282 **Proof Strategy.** Our main aim in this section will be to prove the following theorem:

283 **Theorem 7.** CPS[*failure treewidth*] is in FPT.

284 It is easy to see that failure treewidth is at most the failure cover number plus 1 (consider, *e.g.*, a tree
285 decomposition of the failure graph consisting of a sequence of bags, each containing the algorithms
286 and tests forming the cover and one additional test or algorithm). Hence, once we establish Theorem 7
287 we obtain the following as an immediate corollary:

288 **Corollary 8.** CPS[*failure cover number*] is in FPT.

289 The proof of Theorem 7 is broken into two main steps, and we provide a high-level overview of these
290 below.

291 In the first step, we show that there exists an optimal solution of size at most $\mathcal{O}(tw(G_\mathcal{I}) \cdot \log{(m+n)})$.
292 To argue this property, we consider a balanced separator $B$ of $G_\mathcal{I}$ of size $tw(G_\mathcal{I}) + 1$, and assume
293 w.l.o.g. that, in an optimal solution, an algorithm on the left side of $B$ occurs before an algorithm on
294 the right side of $B$. It can be shown that there exists an optimal solution which behaves as follows:
295 the first algorithm located on the right side of $B$ is followed by at most $tw(G_\mathcal{I}) + 1$ other algorithms.
296 With this insight, we can show that $tw(G_\mathcal{I}) + 3$ many algorithms solve at least a constant fraction of
297 all tests in the instance. Applying this argument recursively allows us to obtain the desired bound on
298 the size of the optimal solution.

299 In the second step, we solve the problem using dynamic programming on a tree decomposition of
300 $G_\mathcal{I}$, by utilizing the upper bound on the solution length derived in the first step. The running time is
301 $\mathcal{O}^*(4^\ell \cdot \ell^{\text{tw}(G_\mathcal{I})} \cdot \text{tw}(G_\mathcal{I})!)$. To make the dynamic programming approach work, for a current bag in
302 the tree decomposition, and for each position in the schedule, we remember whether this position will
303 be filled by an algorithm from the "future" (*i.e.*, has not been seen yet), whether it was already filled
304 in the "past", or whether it will be filled by an algorithm from the current bag. We also remember
305 specifically which algorithm is the "first" from the future, which is the "first" from the past, and what
306 are the positions of the algorithms in the bag. Moreover, for each test in the bag, we remember the
307 position of the algorithm that solves the test.

308 **The Proof.** We start with the following lemma:

309 **Lemma 9.** *There exists a minimum cost schedule for* CASCADING PORTFOLIO SCHEDULING *with*
310 *length at most* $(2tw(G_\mathcal{I}) + 5) \cdot \log(m+n)$.

311 *Proof.* As our starting point, we establish the following claim concerning *proper valid schedules*,
312 which are schedules where each algorithm solves at least one test that has not been solved by previous
313 algorithms.

314 **Claim 10.** *Let* $(A, B)$ *be a separation in* $G_\mathcal{I}$ *with separator* $X = A \cap B$. *Then in every proper valid*
315 *schedule, either all the algorithms from* $A \setminus X$ *or all the algorithms from* $B \setminus X$ *are among the last*
316 $|X \cap T| + 1$ *scheduled algorithms.*

317 *Proof of Claim.* Let $\tau$ be such a valid schedule. Assume, w.l.o.g., that an algorithm $\alpha_A \in A \setminus X$ is
318 scheduled before any algorithm from $B \setminus X$, and let $\alpha_B \in B \setminus X$ be the first algorithm from $B \setminus X$
319 in $\tau$. Since $(A, B)$ is a separation in the failure graph with separator $X$, every algorithm in $A \setminus X$
320 solves every test in $B \setminus X$ and vice versa. Hence, it is easy to see that the only tests that may not
321 have been solved when $\tau$ passes $\alpha_B$ are tests in $X$. Since in $\tau$ every algorithm solves at least one
322 new test, there are at most $|X \cap T|$ algorithms scheduled after $\alpha_B$, and the claim follows. $\qquad\square$

323 Let $\mathcal{I} = (\mathcal{A}, T, \text{cost}, S)$ be an instance of CASCADING PORTFOLIO SCHEDULING. We will prove
324 the lemma by induction on $|\mathcal{A}| + |T|$ and by restricting our attention to proper valid schedules only;
325 the latter is justified by the fact that every valid schedule can be turned into a proper valid schedule
326 without increasing its cost. Clearly, if $|\mathcal{A}| + |T| \leq 2$ then the lemma holds.

327 Now let us assume inductively, for every instance $\mathcal{I}' = (\mathcal{A}', T', \text{cost}', S')$ with $|\mathcal{A}'| + |T'| < |\mathcal{A}| + |T|$,
328 that every proper valid schedule has length at most $(2tw(G_{\mathcal{I}'} + 5) \cdot \log(|\mathcal{A}'| + |T'|)$. Let $\tau$ be such a
329 valid schedule for $\mathcal{I}$, and let $(A, B)$ be a $\frac{2}{3}$-balanced separation in $G_\mathcal{I}$ with separator $X$ of size at
330 most $tw(G_\mathcal{I}) + 1$, it is well known that such separation always exists (see, *e.g.*, [4, Lemma 7.20]).
331 If the length of $\tau$ is at most $2|X| + 3$, then we are done. Otherwise, the sets $A, B, X$ satisfy the
332 conditions of Claim 10, and hence all the algorithms from one of the parts, say $B \setminus X$, are among the
333 last $|X \cap T| + 1$ scheduled algorithms. Moreover, since $\tau$ contains more than $2|X| + 3$ algorithms,
334 there is an algorithm $\alpha_A$ that is the first algorithm from $A \setminus X$ in $\tau$. Now let $\mathcal{I}' = (\mathcal{A}', T', \text{cost}', S')$
335 be the instance obtained from $\mathcal{I}$ by removing:

- $\alpha_A$ and all tests solved by $\alpha_A$,
- all tests and algorithms in $B \setminus X$, and
- all tests solved in $\tau$ by some algorithm in $B \setminus X$ and no algorithm before them in the schedule.

Note that $|\mathcal{A}'| + |T'| \leq \frac{2}{3}(|\mathcal{A}| + |T|)$. Now let $\tau'$ be the schedule obtained from $\tau$ by removing $\alpha_A$, all algorithms from $B$, and all algorithms from $X$ that appear before $\alpha_A$ and only succeed on tests in $B$. Note that we removed at most $1 + |X \cap T| + 1 + |X \cap \mathcal{A}| = |X| + 2$ algorithms from $\tau$. Now, since after $\alpha_A$ all the tests in $B \setminus X$ were solved and every algorithm in $\tau$ solves at least one new test (*i.e.*, a test not solved by previous algorithms), it follows that every algorithm in $\tau'$ solves at least one new test. Moreover, we removed from $T'$ all the tests that were solved in $\tau$ by $\alpha_A$ or an algorithm in $B$ and hence $\tau'$ is a valid schedule for $\mathcal{I}'$. Finally, note that $tw(G_{\mathcal{I}'}) < tw(G_{\mathcal{I}})$. Hence by our assumption the length of $\tau'$ is at most $(2tw(G_{\mathcal{I}}) + 5) \log(|\mathcal{A}'| + |T'|) \leq (2tw(G_{\mathcal{I}}) + 5) \log(\frac{2}{3}(|\mathcal{A}| + |T|)) = (2tw(G_{\mathcal{I}}) + 5) \log(|\mathcal{A}| + |T|) - (2tw(G_{\mathcal{I}}) + 5) \log(\frac{3}{2})$. The lemma then follows by the fact that $tw(G_{\mathcal{I}}) + 3 < (2tw(G_{\mathcal{I}}) + 5) \log(\frac{3}{2})$ for $tw(G_{\mathcal{I}}) \geq 1$. $\qquad\square$

This concludes the first step towards the proof of Theorem 7. For the second step, we need to show the following lemma:

**Lemma 11.** *A minimum cost schedule for* CASCADING PORTFOLIO SCHEDULING *among the schedules of length exactly $\ell$ can be computed in time $\mathcal{O}^*(4^\ell \cdot \ell^{\mathrm{tw}} \cdot \mathrm{tw}!)$.*

*Proof Sketch.* As with virtually all fixed-parameter algorithms parameterized by treewidth, we use leaf-to-root dynamic programming along a nicetree decomposition (in this case of the failure graph $G_{\mathcal{I}}$)—see for instance the numerous examples presented in the literature [5, 4]. However, due to the specific nature of our problem, the records dynamically computed by the program are far from standard. This can already be seen by considering the size of our records: while most such dynamic programming algorithms only store records that have size bounded by a function of the treewidth, in our case the records will also have a polynomial dependence on $m$.

As a starting point, we will use the known algorithm of Bodlaender *et al.* [2] to compute a nicetree-decomposition of width at most $5 \cdot tw(G_{\mathcal{I}})$. We proceed by formalizing the used records. Let $X_i$ be a bag in the tree decomposition. A *configuration* w.r.t. $X_i$ is a tuple $(\alpha_{\mathrm{past}}, \alpha_{\mathrm{future}}, \sigma, \delta)$, where
- $\alpha_{\mathrm{past}}$ is an algorithm that has been forgotten in a descendant of $X_i$,
- $\alpha_{\mathrm{future}}$ is an algorithm that has not been introduced yet in $X_i$,
- $\sigma : [\ell] \to \{ \text{"past", "future"} \} \cup (\mathcal{A} \cap X_i)$, and
- $\delta : T \cap X_i \to [\ell]$.

The interpretation of the configuration is that $\sigma[j]$ tells us whether the $j$-th algorithm in the final schedule has been already introduced and forgotten ("past"), it is a specific algorithm in the bag $X_i$, or it has not been introduced yet ("future"). The function $\delta$, for a test $t$, tells us that $t$ is solved by the $\delta[t]$-th algorithm and not solved by any algorithm before. The entries $\alpha_{\mathrm{past}}$ and $\alpha_{\mathrm{future}}$ represent the specific algorithms with the lowest indices $j, j'$ with $\sigma[j] = $ "past" and $\sigma[j'] = $ "future", respectively. We will refer to the index $j$ such that $\sigma[j] = $ "past" for all $j' < j$ is $\sigma[j'] \neq $ "past" as *the index of the first "past"* and, similarly, to the index $j$ such that $\sigma[j] = $ "future" and for all $j' < j$ is $\sigma[j'] \neq $ "future" as *the index of the first "future"*. Note that the indices of the first "past" and the first "future", respectively, are interpreted as precisely the positions of $\alpha_{\mathrm{past}}$ and $\alpha_{\mathrm{future}}$ in the schedule.

We say that a configuration $C = (\alpha_{\mathrm{past}}, \alpha_{\mathrm{future}}, \sigma, \delta)$ w.r.t. $X_i$ is *admissible* if
- for each algorithm $\alpha \in \mathcal{A} \cap X_i$, there exists precisely one index $j$ such that $\sigma[j] = \alpha$;
- if $\delta[t] = j$, then for every $j' < j$:
  - if $\sigma[j'] \in \mathcal{A} \cap X_i$ then $\sigma[j']$ does not solve $t$, and
  - if $\sigma[j'] = $ "past" (resp. $\sigma[j'] = $ "future"), then $\alpha_{\mathrm{past}}$ (resp. $\alpha_{\mathrm{future}}$) does not solve $t$; and
- if $\delta[t] = j$ and the $j$-th algorithm is determined from $C$ (that is $\sigma[j] \in \mathcal{A} \cap X_i$ or $\sigma[j]$ is the index of the first "past" or "future", respectively), then the $j$-th algorithm defined by $C$ solves $t$.

Note that if we take any valid schedule, we can project it w.r.t. a bag $X_i$ and obtain a configuration $(\alpha_{\mathrm{past}}, \alpha_{\mathrm{future}}, \sigma, \delta)$. Such a configuration will always be admissible and so we can restrict our attention to admissible configurations only.

To simplify the notation we let $\Gamma_i[C] = \infty$ if $C$ is not an admissible configuration w.r.t. $X_i$.

**Claim 12.** *There are at most $2^\ell \cdot \ell^{|X_i|} \cdot |X_i|! \cdot m^2$ admissible configurations.*

*Proof of Claim.* First, there are at most $m^2$ possibilities for the algorithms $\alpha_{\text{past}}$ and $\alpha_{\text{future}}$. Now, let $a, b \in \mathbb{N}$ be such that $a = |X_i \cap \mathcal{A}|$ and $b = |X_i \cap T|$. Note that $a + b = |X_i|$. There are $\binom{\ell}{a} \leq \ell^a$ possible position for the algorithms of $X_i$ in $\sigma$ and for each of these we have $|X_i|!$ possibilities for the specific algorithm to be at the given position. Then we have $2^{\ell-a}$ possibilities for the remaining entries of $\sigma$. Finally, there are $\ell^b$ functions from $X_i \cap T$ to $[\ell]$. Hence there are at most $m^2 \cdot \ell^a \cdot a! \cdot 2^{\ell-a} \cdot \ell^b \leq 2^\ell \cdot \ell^{|X_i|} \cdot |X_i|! \cdot m^2$ admissible configurations. □

Now for each $X_i$, we will compute a table $\Gamma_i$ that contains an entry for each admissible configuration $C$ such that $\Gamma_i[C] \in \mathbb{N}$ is the best cost, w.r.t. configuration $C$, of the already introduced tests restricted to the already introduced algorithms and the algorithm $\alpha_{\text{future}}$.

Clearly, the minimum cost schedule of the instance gives rise to some admissible configuration $C$ w.r.t. the root node $X_r$ of the tree decomposition. Hence $\Gamma_r[C]$ contains the minimum cost of a schedule of length $\ell$. In the remainder of the proof, we show how to update the respective nodes of tree decomposition depending on their children.

**Claim 13.** *If $X_i$ is a leaf node, then $\Gamma_i$ can be computed in $\mathcal{O}(|\Gamma_i|)$ time.*

*Proof of Claim.* Note that $X_i = \emptyset$ and that none of the algorithms has been introduced in any leave node. Let $\sigma_{future}$ denote the function $\sigma_{future}[i] = $ "future" for all $i \in [\ell]$. Then the only admissible configuration looks like $(\emptyset, \alpha, \sigma_{future}, \emptyset)$, where $\alpha \in \mathcal{A}$. Moreover, since no tests or algorithms were introduced at that point, the cost of all of these configurations is zero. □

**Claim 14.** *If $X_i$ is an introduce node for a test with the only child $X_j$, then $\Gamma_i$ can be computed in $\mathcal{O}(|\Gamma_i|)$ time.*

*Proof of Claim.* Let $t$ be the newly introduced test and let $C = (\alpha_{\text{past}}, \alpha_{\text{future}}, \sigma, \delta)$ be an admissible configuration w.r.t. $X_i$. Note that since, $X_j$ is a separator between the forgotten algorithms and $t$, it follows that $\alpha_{\text{past}}$ solves $t$ and hence, because $C$ is admissible, the index $\delta[t]$ of the algorithm that solve the test $t$ has to be smaller or equal to the index of the first "past" in $\sigma$. Hence, it is simple to compute the cost $c_t$ of $t$ w.r.t. all already introduced algorithms and $\alpha_{\text{future}}$. Moreover, this cost is unique and does not depend on forgotten algorithms other than $\alpha_{\text{past}}$. Now, let $C' = (\alpha_{\text{past}}, \alpha_{\text{future}}, \sigma, \delta')$ be the configuration w.r.t. $X_j$, where $\delta'$ is the restriction of $\delta$ missing the entry for $t$. Since $\Gamma_j[C']$ contains the best cost w.r.t. $C'$ and the difference between $C'$ and $C$ and between $X_j$ and $X_i$ respectively is only the test $t$, it is easy to observe that $\Gamma_i[C] = \Gamma_j[C'] + c_t$. □

**Claim 15.** *If $X_i$ is an introduce node for an algorithm with the only child $X_j$, then $\Gamma_i$ can be computed in $\mathcal{O}(|\Gamma_i|)$ time.*

*Proof of Claim.* Let $C = (\alpha_{\text{past}}, \alpha_{\text{future}}, \sigma, \delta)$ be an admissible configuration w.r.t. $X_i$ and let $\alpha$ be the newly introduced algorithm. We compute $\Gamma_i[C]$ as follows. Let $\sigma' : [\ell] \to \{$ "past", "future" $\} \cup (\mathcal{A} \cap X_i$ be the function such that $\sigma'[k] = \sigma'[k]$ if $\sigma[k] \neq \alpha$ and $\sigma'[k] = $ "future" if $\sigma[k] = \alpha$.

We distinguish two cases depending on the position of $\alpha$ in $\sigma$.

First, $\sigma[k] = \alpha$ and there exists $k' \in \mathbb{N}$ such that $k' < k$ and $\sigma[k'] = $ "future". Let $C'$ denote configuration $(\alpha_{\text{past}}, \alpha_{\text{future}}, \sigma', \delta)$. Clearly, $\sigma'$ is admissible w.r.t. $X_j$. Furthermore, all already forgotten tests are solved by $\alpha_{\text{future}}$, which is before $\alpha$ in $\sigma$ and hence their costs is already accounted for in $\Gamma_j[C']$. Therefore, we only need to account for the tests in $X_i$ which are solved either by $\alpha$ or later. Let $k \in \mathbb{N}$ be such that $\sigma[k] = \alpha$. We let $\Gamma_i[C] = \Gamma_j[C'] + \sum_{t \in \{t' \mid \delta[t'] \leq k\}} \text{cost}(\alpha, t)$.

Second, $\sigma[k] = \alpha$ and there does not exist $k' \in \mathbb{N}$ such that $k' < k$ and $\sigma[k'] = $ "future". We let $C'$ denote configuration $(\alpha_{\text{past}}, \alpha, \sigma', \delta)$. Clearly, $\sigma'$ is admissible w.r.t. $X_j$. All already forgotten tests are solved latest by $\alpha$ and their costs is already accounted for in $\Gamma_j[C']$. However, we added new $\alpha_{\text{future}}$ in this case and we have to add the cost of running $\alpha_{\text{future}}$ for the tests in $X_i$ that are solved after the first "future" in $\sigma$. Let $k \in \mathbb{N}$ be such that $\sigma[k] = $ "future" and for all $k' < k$ is $\sigma[k] \neq future$. We let $\Gamma_i[C] = \Gamma_j[C'] + \sum_{t \in \{t' \mid \delta[t'] \leq k\}} \text{cost}(\alpha_{\text{future}}, t)$. □

**Claim 16.** *If $X_i$ is a forget node, which forgets a test $t$, with the only child $X_j$, then $\Gamma_i$ can be computed in $\mathcal{O}(\ell|\Gamma_i|)$ time.*

*Proof of Claim.* Let $C = (\alpha_{\text{past}}, \alpha_{\text{future}}, \sigma, \delta)$ be an admissible configuration w.r.t. $X_i$. Forgetting a test does not change the costs of introduced tests w.r.t. introduced algorithms. Hence, we only need to find a configuration w.r.t. $X_j$ of the lowest cost that after removing $t$ from $\delta$ results in $C$. For $k \in [\ell]$ let $C_k$ be a configuration $(\alpha_{\text{past}}, \alpha_{\text{future}}, \sigma, \delta_k)$ such that $\delta_k[t'] = \delta[t']$ for $t \in (X_i \cap T)$ and $\delta_k[t'] = k$. We let $\Gamma_i[C] = \min_{k \in [\ell]} \Gamma_j[C_k]$. □

**Claim 17.** *If $X_i$ is a forget node, which forgets an algorithm $\alpha$, with the only child $X_j$, then $\Gamma_i$ can be computed in $\mathcal{O}((\ell + m)|\Gamma_i|)$ time.*

*Proof of Claim.* Let $C = (\alpha_{\text{past}}, \alpha_{\text{future}}, \sigma, \delta)$ be an admissible configuration w.r.t. $X_i$. Clearly, when we forget an algorithm, the cost of schedule given by $\sigma$ w.r.t already introduced algorithms and tests does not change. Hence, we just need to choose the best configuration of $X_j$ that can result in $C$. Let $k$ be the lowest index such that $\sigma[k] = $ "past". We distinguish two cases depending on whether $\alpha_{\text{past}} = \alpha$ or not.

First, if $\alpha_{\text{past}} = \alpha$, then for an already forgotten algorithm $\alpha'$ let us denote by $C_{\alpha'}$ the configuration $(\alpha', \alpha_{\text{future}}, \sigma', \delta)$ such that $\sigma'[k] = \alpha$ and for all $k' \neq k$ $\sigma'[k] = \sigma[k]$. In this case we let $\Gamma_i[C] = \min_{\alpha'} \Gamma_j[C_{\alpha'}]$.

If $\alpha_{\text{past}} \neq \alpha$, then for all $k' > k$ such that $\sigma[k'] = $ "past" we let $C_{k'}$ be the configuration $(\alpha_{\text{past}}, \alpha_{\text{future}}, \sigma_{k'}, \delta)$ such that $\sigma'[k'] = \alpha$ and $\sigma'[k''] = \sigma[k'']$ for $k'' \neq k'$. We let $\Gamma_i[C] = \min_{k'} \Gamma_j[C_{k'}]$. $\square$

**Claim 18.** *If $X_i$ is a join node with children $X_{j_1}$ and $X_{j_2}$, then $\Gamma_i$ can be computed from $\Gamma_{j_1}$ and $\Gamma_{j_2}$ in $\mathcal{O}(2^\ell m|\Gamma_i|)$ time.*

*Proof of Claim.* Let $C = (\alpha_{\text{past}}, \alpha_{\text{future}}, \sigma, \delta)$ be an admissible configuration w.r.t. $X_i$. In this case, we need to go through all the possibilities how $C$ can decompose into two admissible configurations $C_{j_1} = (\alpha^1_{\text{past}}, \alpha^1_{\text{future}}, \sigma^1, \delta^1)$ and $C_{j_2} = (\alpha^2_{\text{past}}, \alpha^2_{\text{future}}, \sigma^2, \delta^2)$ w.r.t. $X_{j_1}$ and $X_{j_2}$, respectively. We first enumerate the necessary conditions for such two configurations.

1. $\delta = \delta^1 = \delta^2$,

2. $\sigma, \sigma^1$, and $\sigma^2$ agree on the position of the algorithms in $X_i$,

3. if $\sigma[k] = $ "future", then necessarily $\sigma^1[k] = \sigma^2[k] = $ "future",

4. if $\sigma[k] = $ "past" then either $\sigma^1[k] = $ "past" and $\sigma^2[k] = $ "future" or $\sigma^1[k] = $ "future" and $\sigma^2[k] = $ "past",

5. $\alpha_{\text{past}}$ has to be equal to either $\alpha^1_{\text{past}}$ or $\alpha^2_{\text{past}}$, and

6. $\alpha^1_{\text{future}}$ is either $\alpha_{\text{future}}$ or $\alpha^2_{\text{past}}$ and $\alpha^2_{\text{future}}$ is either $\alpha_{\text{future}}$ or $\alpha^1_{\text{past}}$.

It is straightforward to see that there are at most $2^\ell + 2 \cdot m$ such $(C_{j_1}, C_{j_2})$ pairs. Having such a pair and costs $\Gamma_{j_1}[C_{j_1}], \Gamma_{j_2}[C_{j_2}]$, we show how to compute the cost of $C$ if it rises from the combination of $C_{j_1}$ and $C_{j_2}$. Afterwards, we just need to go through all the combinations and pick the one that gives the lowest possible cost.

First note that the sets of tests that are forgotten in $X_{j_1}$ and $X_{j_2}$ are disjoin,t and the cost of these tests is included in $\Gamma_{j_1}[C_{j_1}]$ and $\Gamma_{j_2}[C_{j_2}]$ respectively and this cost will not change after joining these to configurations. This is true since the only not introduced algorithms, in the respective bags, that will run on these tests are $\alpha^1_{\text{future}}$ and $\alpha^2_{\text{future}}$, respectively, and their costs are already included in $\Gamma_{j_1}[C_{j_1}]$ and $\Gamma_{j_2}[C_{j_2}]$. Hence, if $X_i$ does not contain any test, then the cost of combining these two configurations is precisely $\Gamma_{j_1}[C_{j_1}] + \Gamma_{j_2}[C_{j_2}]$. Similarly, for a test $t \in X_i$ and a forgotten algorithm other than $\alpha^1_{\text{past}}$ or $\alpha^2_{\text{past}}$ (which could be $\alpha^2_{\text{future}}$ or $\alpha^1_{\text{future}}$, respectively) the cost of running this algorithm is already counted in precisely one of $\Gamma_{j_1}[C_{j_1}]$ and $\Gamma_{j_2}[C_{j_2}]$. Hence, the only thing that can be counted twice in $\Gamma_{j_1}[C_{j_1}] + \Gamma_{j_2}[C_{j_2}]$ is the cost of running a test in $X_i$ w.r.t. algorithm in $X_i \cup \{\alpha_{\text{future}}, \alpha^1_{\text{future}}, \alpha^2_{\text{future}}\}$, which can be easily checked, computed from $\sigma$'s and $\delta$, and afterwards subtracted from $\Gamma_{j_1}[C_{j_1}] + \Gamma_{j_2}[C_{j_2}]$. Finally, if none of $\alpha^1_{\text{future}}$ and $\alpha^2_{\text{future}}$ is equal to $\alpha_{\text{future}}$, we need to add to the final cost the cost of running $\alpha_{\text{future}}$ on tests that are in $X_i$ and are solved at least at the position of the first "future" in $\sigma$ or later (*e.g.*, $\delta[t] \geq k$, where $k$ is the position of the first "future" in $\sigma$). This conclude the computation of the costs of combining two configurations from children in join node and by taking the minimum among all such combinations also the proof of the claim. $\square$

To conclude, the last four claims show that it is possible to dynamically compute our records from the leaves of a nice tree decomposition to its root; once the records are known for the root, the algorithm has all the information it needs to output with the solution. $\square$

Since, by Lemma 9 the length of a minimum cost schedule is at most $(2tw(G_\mathcal{I}) + 5) \cdot \log(n + m)$, by applying the above algorithm for each length between 1 and $(2tw(G_\mathcal{I}) + 5) \cdot \log(n + m)$ we get a runtime of $\mathcal{O}^*(4^{(2tw(G_\mathcal{I})+5)\cdot\log(n+m)} \cdot ((2tw(G_\mathcal{I}) + 5) \cdot \log(n + m))^{tw(G_\mathcal{I})} \cdot tw(G_\mathcal{I})!) = \mathcal{O}^*(4^{2tw(G_\mathcal{I})+5} \cdot (2tw(G_\mathcal{I}) + 5)^{tw} \cdot \log(n + m)^{tw(G_\mathcal{I})} \cdot tw(G_\mathcal{I})!)$. It is well known [20] that, for

a parameter $k$ and input size $N$, a running time of the form $\mathcal{O}^*((\log(N))^k)$ is FPT. It follows that CPS[*failure treewidth*] is fixed-parameter tractable, and the proof of Theorem 7 is complete.

## 5.2 Using the Success Cover Number

The aim of this section is to establish the fixed-parameter tractability of CPS[*success cover number*], which can be viewed as a dual result to Corollary 8. The techniques used to obtain this result are entirely different from those used in the previous subsection; in particular, the proof is based on a significant extension of the ideas introduced in the proof of Proposition 1.

**Theorem 19.** CPS[*success cover number*] is in FPT.

*Proof.* Let $\mathcal{I}$ be an instance of CPS[$\text{cov}_s$]. Our first step is to compute a witness for the success cover number $\text{cov}_s$, *i.e.*, a set of algorithms $\mathcal{A}'$ and tests $T'$ such that $|\mathcal{A}' \cup T'| = \text{cov}_s$ and each pair in $S$ has a non-empty intersection with $\mathcal{A}' \cup T'$; as discussed in Subsection 2, this can be done in polynomial time [8, Proposition 1]. Let $V = 2^{\mathcal{A}' \cup T'}$ be the set of all subsets of $\text{cov}_s$. We will construct a directed arc-weighted graph $D$ with vertex set $V \cup \{x\}$, and with the property that each shortest path from $\emptyset$ to $x$ precisely corresponds to a minimum-cost schedule for the input instance $\mathcal{I}$. Intuitively, reaching a vertex $v$ in $D$ which corresponds to a certain set of algorithms $\mathcal{A}_0$ and tests $T_0$ means that the schedule currently contains the algorithms in $\mathcal{A}_0$ plus an optimal choice of algorithms which can process the remaining tests in $T_0$; information about the ordering inside the schedule is not encoded by the vertex $v$ itself, but rather by the path from $\emptyset$ to $v$.

In order to implement this idea, we will add the following arcs to $D$. To simplify the description, let $\mathcal{A}^*$ be an arbitrary subset of $\mathcal{A}'$ and $T^*$ be an arbitrary subset of $T'$. First of all, for each $\mathcal{A}^*$ such that for every test $t \in T \setminus T'$ there is some $\alpha \in \mathcal{A}^*$ satisfying $(\alpha, t) \in S$, we add the arc $(\mathcal{A}^* \cup T', x)$ and assign it a weight of 0. This is done to indicate that $\mathcal{A}^* \cup T'$ corresponds to a valid schedule.

Second, for each $\mathcal{A}^*$ that is a proper subset of $\mathcal{A}'$, $\alpha_0 \in \mathcal{A}' \setminus \mathcal{A}^*$, and $T^*$, we add the arc $e$ from $\mathcal{A}^* \cup T^*$ to $\mathcal{A}^* \cup \{\alpha_0\} \cup T^* \cup T_0$, where $T_0$ contains every test $t_0 \in T'$ such that $(\alpha_0, t_0) \in S$. In order to compute the weight of this arc $e$, we first compute the set $T_e$ of all tests outside of $T^*$ where $\alpha_0$ will be queried (assuming $\alpha_0$ is added to the schedule at this point); formally, $t \in T_e$ if $t \notin T^*$ and for each $\alpha' \in \mathcal{A}^*$ it holds that $(\alpha', t) \notin S$. For clarity, observe that $T_0 \subseteq T_e$. Now, we set the weight of $e$ to $\sum_{t \in T_e} \text{cost}(\alpha_0, t)$.

To add our third and final set of edges, we first pre-compute for each $T_\lambda \subseteq T' \setminus T^*$ an algorithm $\alpha_\lambda \in \mathcal{A} \setminus \mathcal{A}'$ such that:

1. for each $t_\lambda \notin T^*$, $(\alpha_\lambda, t_\lambda) \in S$ iff $t_\lambda \in T_\lambda$ (*i.e.*, $\alpha_\lambda$ successfully solves exactly $T_\lambda$), and

2. among all possible algorithms satisfying the above condition, $\alpha_\lambda$ achieves the *minimum cost* for all as-of-yet-unprocessed tests. Formally, $\alpha_\lambda$ minimizes the term $\text{price}(\alpha_\lambda) = \left(\sum_{t \in (T' \setminus T^*)} \text{cost}(\alpha_\lambda, t)\right) + \left(\sum_{t \notin T' : \forall \alpha \in \mathcal{A}^* : (\alpha, t) \notin S} \text{cost}(\alpha_\lambda, t)\right)$.

Now, we add an arc $e$ from each $\mathcal{A}^* \cup T^*$ to each $\mathcal{A}^* \cup T^* \cup T_\lambda$, where $T_\lambda$ is defined as above and associated with the test $\alpha_\lambda$. The weight of $e$ is precisely the value $\text{price}(\alpha_\lambda)$.

Note that since the graph $D$ has $2^{\text{cov}_s} + 1$ many vertices, a shortest path $P$ from $\emptyset$ to $x$ in $D$ can be computed in time $2^{\mathcal{O}(\text{cov}_s)}$. Moreover, it is easy to verify that $D$ can be constructed from an instance $\mathcal{I}$ in time at most $2^{\mathcal{O}(\text{cov}_s)} \cdot |\mathcal{I}|^2$. At this point, it remains to verify that a shortest $\emptyset$-$x$ path $P$ in $D$ can be used to obtain a solution for $\mathcal{I}$.

Consider such a path $P$, and let $Q$ be the schedule constructed iteratively as follows. At the beginning we set $Q = \emptyset$. We will follow the path $P$ from $\emptyset$ to $x$ and add an algorithm to $Q$ whenever $P$ traverses an arc with non-zero weight. In particular, whenever $P$ uses an arc from some $\mathcal{A}^* \cup T^*$ to some $\mathcal{A}^* \cup \{\alpha_0\} \cup T^* \cup T_0$ where $\alpha_0 \in \mathcal{A}'$ (*i.e.*, an arc from our second group of created edges), we add $\alpha_0$ to the end of $Q$. Similarly, whenever $P$ uses an arc from some $\mathcal{A}^* \cup T^*$ to some $\mathcal{A}^* \cup T^* \cup T_\lambda$, we add the algorithm $\alpha_\lambda$ to the end of $Q$. It is easy to verify that at each point of our iterative construction of $Q$, the cost of an arc used by $P$ is precisely the same as the processing cost the algorithm that was last added to $Q$ spends on all tests which remain unsolved at this point. Moreover, by the construction of our first group of arcs, once $P$ reaches $x$ we end up with a valid schedule $Q$. It follows that $Q$ is a valid schedule with cost equal to the weight of $P$.

To conclude the proof, among all valid schedules of minimum cost, let $Q'$ be one such schedule with the minimum weight. It now suffices to show that there exists a path $P'$ with the same weight as

547 the cost of $Q'$. To find this path $P'$, we will be essentially be reversing the arguments made in the
548 previous paragraph. Notably, we will find $P'$ by choosing the $i$-th arc to follow based on the $i$-th
549 algorithm in $Q'$. As our inductive assumption, we will moreover claim that:

    1. if the $i$-th node we reach in $P'$ is $\mathcal{A}^* \cup T^*$, then the first $i$ algorithms in $Q'$ contain $\mathcal{A}^*$ and
    that the algorithms in $Q'$ solve precisely those tests in $T'$ which lie in $T^*$; and

    2. the weight of $P'$ up to the $i$-th node is precisely equal to the cost of processing all tests by
    the first $i$ algorithms in $Q'$.

554 This inductive assumption is clearly true at position $i = 0$, since $P'$ starts at $\emptyset$. So, assume the
555 inductive assumption holds at some position $i$, the vertex reached by $P'$ after taking the $i$-th arc is
556 some $\mathcal{A}^* \cup T^*$, and the $(i+1)$-st algorithm in $Q'$ is some $\alpha_j$.

557 If $\alpha_j \in \mathcal{A}'$, then we make our path $P'$ follow the arc to $\mathcal{A}^* \cup \{\alpha_j\} \cup T^* \cup T_0$, where $T_0$ contains
558 every test $t_0 \in T'$ such that $(\alpha_j, t_0) \in S$; such an arc can be found among the arcs created in our
559 second group, the set $T_0$ is constructed in a way which preserves our first inductive assumption, and
560 the cost of this arc will precisely match the cost of processing all remaining tests by $\alpha_j$. Hence, both
561 inductive assumptions will remain satisfied.

562 On the other hand, if $\alpha_j \notin \mathcal{A}'$, then due to the optimality of $Q'$ we can assume that there exists
563 some non-empty set $T_\lambda \subseteq T' \setminus T^*$ of tests solved by $\alpha_j$, *i.e.*, $t \in T_\lambda$ iff $(\alpha_j, t) \in S$. To be more
564 precise, observe that an $\alpha_j$ outside of $\mathcal{A}'$ can only successfully process tests in $T'$, and if such $\alpha_j$
565 does not succeed on any as-of-yet remaining test we may simply remove $\alpha_j$ from $Q'$, contradicting
566 our assumption about the (length) minimality of $Q'$. Moreover, among *all* algorithms which succeed
567 precisely on $T_0$, we note that $\alpha_j$ must run with the minimum cost for all test instances that remain
568 unsolved as of this point—in other words, $\alpha_j$ must be an algorithm with minimum price, as defined
569 when constructing our third set of edges. Indeed, if this were not the case, we could replace $\alpha_j$ with
570 the algorithm $\alpha_\lambda$ pre-computed for $T_\lambda$, leading to a schedule $Q'$ of strictly smaller cost. But since
571 $\alpha_j$ is an algorithm with minimum price, there must exist an arc from $\mathcal{A}^* \cup T^*$ to $\mathcal{A}^* \cup T^* \cup T_\lambda$ of
572 cost precisely price($\alpha_j$), and by having $P'$ use this arc we ensure that both inductive assumptions
573 remain valid (the first assumption follows from the definition of $T_\lambda$, and the second follows from the
574 optimality of the price of $\alpha_j$).

575 To summarize, we have shown that for each valid optimal schedule $Q'$ as defined above, there is a
576 corresponding $\emptyset$-$x$ path $P'$ in our graph $D$, and that for each $\emptyset$-$x$ path $P$ in $D$ there is a corresponding
577 valid schedule. These two facts together establish the correctness of our algorithm. $\qquad\square$

## 6 Conclusion

579 We studied the parameterized complexity of the CASCADING PORTFOLIO SCHEDULING problem
580 under various parameters. We identified several settings where the NP-hardness of the problem can
581 be circumvented via exact fixed-parameter algorithms, including cases where (i) the algorithms have
582 a small failure degree, (ii) the tests have a small failure degree, (iii) the evaluation matrix has a small
583 failure cover, and (iv) the evaluation matrix has a small success cover. The first three cases can be
584 seen as settings in which most algorithms succeed on most of the tests, whereas case (iv) can be seen
585 as a setting where most algorithms fail.

586 We have complemented our algorithmic results with hardness results which allowed us to draw a
587 detailed complexity landscape of the problem. We would like to point out that all our hardness results
588 hold even when all costs are unit costs. This finding is significant, as it reveals that the complexity of
589 the problem mainly depends on the success relation and not on the cost mapping.

590 For future work, it would be interesting to extend our study to the more complex setting where up to
591 $p$ algorithms from the portfolio can be run in parallel. Here, the number $p$ could be seen as a natural
592 additional parameter.

## Footnotes

[1]Fixed-parameter tractability is a relaxation of polynomial tractability; definitions are provided in Section 2.