[Reviews · NeurIPS 2019]

Reviewer 1



Post-rebuttal: I am satisfied with the response. I hope the authors can include the QBF data in the final version of this paper. -------------------------- ORIGINALITY ============ The work's main originality is in introducing the study of CPS as a parameterized computational problem. The techniques used to establish fixed-parameter tractability or intractability are fairly standard in the area and cannot be considered new contributions in my opinion. QUALITY & CLARITY ================= The paper systematically analyzes the following parameterizations: * number of algorithms * number of tests * length of the optimal schedule * algorithm and test failure degree * algorithm and test success degree * failure cover number, and more generally, treewidth of failure graph * success cover number The proof sketches in the main submission, and the full proofs in the supplement, are very clear and well-written. The proofs follow standard techniques and are all correct (to the best of my judgement). Where the paper severely lacks, especially as a submission to NeurIPS, is in justifying that any of the above parameterizations are relevant to applications in machine learning and artificial intelligence. Can we expect the failure graph in QBF solving or ImageNet classification models to have a small treewidth or have a small success cover number? SIGNIFICANCE ============ I have mixed feelings about this submission. On the one hand, CPS captures a basic problem in software engineering, and the work establishes some fundamental results about the complexity of the problem. On the other hand, it is going to be of limited interest to the NeurIPS community because it's not clear how the parameterizations relate to actual instances. Also, the submission would be significantly strengthened if the paper looked at the hardness of the specific scenarios 1, 2 and 3.

Reviewer 2



Originality: As far as I can tell, the paper uses fairly standard arguments to achieve its theoretical results. However, I would argue that the simplicity here is a virtue, not a fault. In a sense, the paper is a showcase of the powerful FPT toolkit being applied to the cascading portfolio scheduling question. Quality: The paper thoroughly considers many possible parameterizations of a very general problem (see above). This problem captures many plausible scenarios of how one might want to try running a sequence of algorithms, and encompasses algorithms with unreliable running time or output. Despite the generality of the problem, the paper still manages to establish positive results for a number of cases. I think a reasonable case for acceptance could be made even without the last set of results (failure/success cover number); differentiating between the tractable parameters and intractable ones of (i) and (ii) is already an interesting contribution. Clarity: The paper is well-organized, presenting the parameters in the natural (simple to complex) order. It might be helpful to add a chart to the introduction, indicating the map from parameter to tractable/intractable. Figure 1 was helpful in illustrating the parameters based on covering number, thanks. As one minor nit, the paper said that it would use a spade to mark statements with omitted proofs, but they were not actually marked with spades. Significance: This is a nice foundation to build future work on top of. I think the biggest weakness of this paper is addressing whether these parameters are actually small in any of the original motivators for the problem, i.e. are these algorithms useful for the original questions that lead us here. Leaving this question unanswered is why I would argue that this paper is just an accept instead of a strong accept. Contrast these results with standard treewidth results, where we know that many graphs of interest (e.g. road networks, social networks) have small treewidth, making treewidth a meaningful characterization, not just a characterization that we can prove results for. Is there a good reason to expect failure treewidth to be small? Rebuttal: I've read your rebuttal. Sounds good, I hope your QBF data works out!

Reviewer 3



[Originality] The authors present a parameterized complexity analysis of CPS, which seems to be motivated by the recent success on Matrix Completion. The results systematically explain the tractability and the hardness when CPS is parameterized by # algorithms, # tests, length, algorithm failure/success degree, test failure/success degree, failure/success cover number, which are extensive and clearly original. [Quality] * This paper demonstrates not just positive side (i.e., fixed-parameter tractability) but also negative side (e.g., W[2]-hardness), which strongly reveals when (and why) the exact solution of CPS is hard or easy to find, while the intuitive statement that the success relation governs the computational complexity is convincing. Some of the proof techniques are technically complicated as far as I have read in the main paper though based on fundamental paradigm (such as DP). [Clarity] * I was confused with the proof of Theorem 7: in the deformation from Line 337 to Line 338 in page 7, the authors removed the "\log(n+m)" on the "4" in the runtime; however, the factor 4^{(2tw+5) log(n+m)} seems to be poly(n+m)^(2tw+5) for some polynomial function and it is not FPT (but XP) if so. Please clarify the deformation from Line 337 to Line 338. * Also, including the concrete deformation from O^*((log(N))^k) to O(f(k)+n) for some k in Line 339 (instead of just referring to [19]) makes the paper more self-contained. [Significance] Overall, I thought this is a good paper devoted to the systematic complexity analysis for FPT of the CPS problem, which is motivated by applications. [Minor issues] In proof of Proposition 1, $t$ is used to denote test and destination of the DAG (e.g., Line 155 and Line 158), which should be revised. ==== UPDATE ==== Thank you for providing the feedback. The modification to DP tables makes sense and the fixed-parameter tractability of CPS[failure treewidth] sounds correct.

[Author Response · NeurIPS 2019]

**Response to Reviewer 1**

*"...Provide some evidence that the parameterizations relate to structural properties of failure graphs arising from real applications"*

There are many application scenarios where our parameterizations are reasonable. For instance, if the failure graph depends on the timeout given by the user, then the user has some control over its density, and therefore can tune the parameters. However, if this data is fine-tuned by the user as part of an experimental evaluation, then this might be subject to the criticism of having fine-tuned the data to reach a positive outcome. Therefore, the data source should be independent from the user, and we are currently reviewing some external data obtained from the QBF setting (where we have no control over the failure rates) for inclusion in the final or journal version of the paper.

*"Analyze the complexity of these problems in the more common scenarios 1, 2, and 3. Do the problems remain NP-hard (without parameterization)? What about parameterized with respect to length of the optimal scheduling?"*

It is not difficult to show that the problem remains NP-hard in all 3 scenarios via a reduction from Min-Sum Set Cover (reference [6] in the paper). As for the parameterized complexity of the problem parameterized by the length of an optimal schedule, it remains W[2]-hard in all 3 scenarios via a reduction from Set Cover; this reduction is the same as the one used for the more general problem. We will add a specific remark about the above into the final version.

**Response to Reviewer 2**

*"the paper said that it would use a spade to mark statements with omitted proofs, but they were not actually marked with spades."*

We apologize for this confusion. We will fix this in the final version.

*"It might be helpful to add a chart to the introduction, indicating the map from parameter to tractable/intractable"*

Thank you for this suggestion. We will add the suggested chart to the introduction.

*"As stated earlier in the review, this submission could be a strong accept if it gave some empirical evidence that its algorithms help with the original questions. Are these parameters actually small?"*

We believe that our main contribution is an in-depth complexity classification of the problems under consideration. As for the size of the parameters in practice, please see our response to Reviewer 1.

**Response to Reviewer 3**

We would like to profoundly thank the reviewer for spotting the issue concerning the running times in Theorem 7, and we apologize for missing this in our proofreading. Fortunately, it is easy to fix the issue by slightly modifying the records kept in the dynamic programming table: while these currently store the exact positions of "important algorithms" w.r.t. the whole schedule of length $\ell$ (which is where the dependency on $\ell$ came from), it suffices to store their relative positions (which can be kept as a permutation of only these algorithms).

More precisely, to fix this issue, the dynamic programming table in Lemma 11 still stores records with the same structure $(\alpha_{\text{past}}, \alpha_{\text{future}}, \sigma, \delta)$ as before, with the difference being that now $\sigma$ captures only the relative positions of the following algorithms (as opposed to their global positions):

    1. the algorithms identified by $\alpha_{\text{past}}$ and $\alpha_{\text{future}}$,

    2. the algorithms in the current bag, and

    3. the algorithms that will solve the tests in the bag.

$\delta$ then only tells us whether the test will eventually be solved by an algorithm in the future or whether it is solved by an already-introduced algorithm (this is needed for processing the *join nodes*). Since in the proofs of Claims 13-18 (in the full version) we only need the relative positions of the algorithms in the bag in order to update the records and their associated costs, all the arguments in these claims seamlessly go through. The outcome is a fixed-parameter algorithm for this case with a runtime of $\mathcal{O}^*(4^{\text{tw}} \cdot \text{tw}^{\text{tw}})$.

We have already prepared a formal write-up of this adapted proof and will make the necessary updates to the final version. (The NeurIPS guidelines explicitly forbid us from linking to a modified version of the paper.)

*"In proof of Proposition 1, $t$ is used to denote test and destination of the DAG..."*.

We will use a different notation for the destination vertex.

[Meta-Review · NeurIPS 2019]

The CPS problem covered in this paper is a basic one in software engineering, and paper establishes some fundamental complexity results about the problem. Even though it is solid work, the authors may find that only a fraction of the NeurIPS community is interested in the problem.